# Identifying suitable tester for evaluating *Striga* resistant lines using DArTseq markers and agronomic traits

Degife Zebire[1,2,3], Abebe Menkir[1]*, Victor Adetimirin[4], Wende Mengesha[1], Silvestro Meseka[1], Melaku Gedil[1]

**1** International Institute of Tropical Agriculture (IITA), Ibadan, Nigeria, **2** Pan Africa University of Life and Earth Sciences Institute, University of Ibadan, Ibadan, Nigeria, **3** Department of Plant Science, College of Agricultural Sciences, Arba Minch University, Arba Minch, Ethiopia, **4** Department of Agronomy, University of Ibadan, Ibadan, Nigeria

* a.menkir@cgiar.org

**Data Availability Statement:** The minimal data set necessary to replicate the study findings is available in the Supporting information and on Dryad (DOI: 10.5061/dryad.zs7h44j89).

## Abstract

A desirable tester that elicits greater genetic difference in *Striga* resistance among test crosses in a breeding program has not been reported. Therefore, this study was conducted to characterize 30 *Striga* resistant yellow endosperm maize inbred lines and three testers with varying resistance levels to *Striga* using DArTseq SNP markers and agronomic traits to identify a suitable tester for resistance hybrid breeding. Marker-based and agronomic trait-based genetic distances were estimated for yellow endosperm maize inbred lines and testers with varying resistance levels to *Striga*. The Marker-based cluster analysis separated the *Striga* resistant lines and testers into two distinct groups. Although the susceptible tester (T3) was the most distantly related to the 30 *Striga* resistant inbred lines, it exhibited a narrower range in genetic distance estimates and poor agronomic performance under *Striga* infestation in crosses with the resistant lines. In contrast, the resistant tester (T2) showed a broader range in genetic distance estimates in pairs with the 30 resistant lines. Also, it formed many high yielding hybrids with desirable traits under parasite pressure. Furthermore, the most significant positive association between agronomic trait-based and marker-based distance estimates (r = 0.389, P = 0.01) was observed when T2 has paired with the *Striga* resistant maize inbred lines. It thus appears that T2 may be used as a suitable tester to determine the breeding value of lines in hybrid maize resistance breeding programs. T2 was the most suitable tester, with a tolerant tester (T1) as an alternative tester to characterize the combining ability of *Striga* resistant maize inbred lines. This result can also encourage other breeders to investigate testers relative discriminating ability with varying levels of resistance in hybrid breeding for resistance to diseases, pests, and other parasitic plants.

## Introduction

Maize (Zea mays L; 2n = 2x = 20) is one of the most important cereal crops worldwide, serving as food, feed and bioenergy [1, 2]. In sub-Saharan Africa (SSA), maize is cultivated in 25 million ha of land mainly by smallholder farmers, producing 38 million metric tons primarily for

**Funding:** This report was part of PhD research of the first author, funded by African Union and the Bill and Malinda Gates Foundation (Grant number: OPP1134248), under the framework of Pan African University and the Stress Tolerant Maize for Africa (STMA), respectively.

**Competing interests:** The authors have declared that no competing interests exist.

**Abbreviations:** CIMMYT, International Maize and Wheat Improvement Center; DArT, Diversity Array technology; DNA, Deoxyribose nuclease; GD, Genetic distance; gDNA, Genomic Deoxyribose nuclease; IITA, International Institute of Tropical Agriculture; L, lines; PCoA, Principal coordinate analysis; PIC, Polymorphic information content; SAS, Statistical Analysis System; SNP, Single Nucleotide Polymorphism; SSA, Sub-saharan Africa; T, Testers; UPGMA, unweighted pair group method with arithmetic mean.

food [3, 4]. However, low maize yields are pervasive in farmers' fields in SSA compared to other parts of the world due to biotic and abiotic stresses hampering its productivity [5, 6]. Amongst the biotic stresses, a parasitic weed known as *Striga hermonthica* (Del.) Benth causes the most damage to cereals, including maize [7], by depleting nutrients and water from the host plants [8]. Intrigued by the potential that hybrids could provide as a catalyst in developing and delivering good quality seed to farmers, considerable efforts have been made at the International Institute of Tropical Agriculture (IITA) to develop *Striga* resistant hybrids for *Striga* affected areas [9]. Inbred lines that combine high yield with resistance to *S. hermonthica* can be used as parents to develop maize hybrids with high grain yields, less *Striga* damage symptoms and a reduced number of emerged parasites. Furthermore, grain yields in hybrids can be increased by improving yield-related traits, including plant height, the number of ears per plant, reduction in the anthesis-silking interval and better ear aspect [9, 10].

Maize breeders at IITA used diverse resistance sources to develop yellow and white maize inbred lines expressing varying field resistance levels to *S. hermonthica* [7, 11, 12]. In this breeding program, a *Striga* resistant inbred tester has been commonly used to evaluate the combining ability of lines to select potential parents for developing resistant hybrids. However, it has been recognized that no single tester may completely fulfil the requirement of being the best in generating resistant hybrids. Considering the uncertainty associated with the choice of the most appropriate tester in hybrid breeding programs [13], understanding the relative value of using other types of testers in eliciting differences among new *Striga* resistant lines is vital in resistance hybrid breeding programs [14–16]. The line x tester mating scheme has been commonly used to generate reliable information about the genetic merit of lines and testers [17–20]. Smith [21] and Hallauer et al. [13] concluded that a homozygous recessive inbred line with a low frequency of desirable dominant genes and favourable alleles would be the most effective tester to identify new inbred lines with a high frequency of favourable alleles in test crosses. However, different testers have been effective in evaluating lines in hybrid maize breeding programs [13, 22]. Mwimali et al. [23] found two *Busseola fusca* resistant single-crosses as desirable testers to detect large genetic differences in resistance among hybrids but highlighted the two testers inability in defining the heterotic orientation of many lines as a significant challenge. Guimaraes et al. [24] recommend the need for continual reassessment of the choice of testers through studies to select the best tester that differentiates resistance reactions among new inbred lines in breeding programs efficiently. It thus appears that identifying the most appropriate testers from among inbred lines with varying levels of resistance to *S. hermonthica* will be useful for the selection of superior *Striga* resistant parental lines of hybrids.

An ideal tester elicits a greater expression of genetic difference among test crosses in a breeding program [25]. However, few studies have been conducted to identify desirable testers for *Striga* resistance hybrid breeding using molecular markers and agronomic traits recorded in hybrids under parasite infestation. The use of such data sets for genetic diversity analyses may provide invaluable information to choose robust testers for evaluating new maize inbred lines as potential parents with diverse genetic backgrounds to develop superior hybrids and source populations of new and diverse maize inbred lines [26]. Selecting a suitable tester using molecular data and hybrid performance may also allow accurate classification of the lines into groups to maximize genetic gain in productivity [27–29].

Morphological traits and molecular markers have been simultaneously used to identify parental lines with diverse genetic backgrounds for developing source populations and superior hybrids [12, 30]. However, the genetic diversity estimates obtained from the two data sets have not been used to identify the most desirable testers for assessing the combining ability of *Striga* resistant maize inbred lines. With advances made in marker technology and the falling costs of DNA sequencing, the diversity array technology (DArT) marker system has become a

cheap, easy and efficient genotyping-by-sequencing platform allowing genome-wide marker discovery efficient analysis of germplasm [31, 32]. DArTseq has been optimized and successfully used for genetic diversity analyses of many crops, including rice [33], cassava [34], pigeon pea [35] and maize [36]. Some inbred lines have been developed from two biparental crosses of elite *Striga* resistant yellow endosperm maize inbred lines, synthetic, an early composite and late-maturing experimental variety [9, 11]. However, no reports have been published on the combined use of DArT markers and agronomic traits for identifying testers to assess the resistance or susceptibility reactions to *S. hermonthica* of the new yellow endosperm maize inbred lines. Such information will be useful in speeding up the development of *Striga* resistant hybrids for cultivation in areas affected by the parasite.

Therefore, the present study is conducted to characterize *Striga* resistant yellow endosperm maize inbred lines and testers with varying resistance levels to *Striga* using DArTseq SNP markers and agronomic traits recorded under parasite pressure to identify the best tester for resistance breeding.

## Materials and methods

### Genetic materials

The germplasm used in this study was developed at IITA, Ibadan, Nigeria. These consisted of thirty yellow endosperm *Striga* resistant maize inbred lines and three testers with varying levels of resistance to *Striga*. Pedigree of the inbred lines and testers were described by Zebire et al. [37]. Lines were designated as L1-L30, and testers presented as T1, T2 and T3 for the tolerant, resistant and susceptible tester, respectively. The 30 inbred lines each were crossed to the three testers to generate 90 test crosses, which were evaluated along with two checks having known tolerant and susceptible reactions to the parasite under *Striga* infested and non-infested conditions at Abuja and Mokwa in Nigeria for two years. All procedures used to precondition *Striga* seed, *Striga* infestation and data collection for field trail were described by Zebire et al. [37].

**DNA extraction and genotyping.** Young growing leaves at the 3 to 4 leaf stage were collected from maize plants grown on the field for DNA extraction. Leaf samples were collected from 4 to 15 leaves of each line and tester. The samples were stored in a deep freezer at -80˚c. Each sample was dried in a Labconco Freezone 2.5L system lyophilizer (Marshall Scientific, USA) before genomic DNA extraction. The extraction of the genomic DNA (gDNA) was carried out according to the DArT protocol (www.diversityarrays.com/files/DArT_DNA_isolation.pdf). The quality and quantity of the DNA were checked by gel electrophoresis using 0.8% agarose gel and NANODROP® spectrophotometer (Thermo Fisher Scientific Inc., Denver, CO, USA). The samples were sent to the Diversity Array technology company [38]. All protocols, including Library construction, sequencing, and SNP calling, were performed at the Diversity Arrays facility, Canberra, Australia. ApeK1 restriction enzyme was used to digest the gDNA, and genotyping-by-sequencing (GBS) libraries were constructed in 96-plex for the samples and sequenced on Illumina HiSeq2000. Raw flow cell output was processed to genotype calls using the trait analysis by association, evolution and linkage (TASSEL)-GBS pipeline. The information of reads and tags found in each sequencing result was aligned to the *Zea mays* L. genome reference, version *AGPV4* (B73 RefGen v4 assembly).

### Data analysis

**Marker-based genetic diversity analysis.** A total of 27,874 SNP markers generated from the present maize panel was received from the DArTseq platform. Quality control was performed to retain only bi-allelic sites, and the SNPs were further filtered using the TASSEL software [39] to maintain only polymorphic SNPs with a maximum of 10% missing values and a

minimum and maximum allele frequency of 0.05 and 0.95, respectively. The final filtered data comprised 6081 SNP markers spanning the ten chromosomes of maize matched the quality criteria and were used for further analysis. Markers were used to calculate polymorphic information content (PIC), minor allele frequency, the number of alleles, gene diversity and heterozygosity using powerMarker software 3.25 version [40]. Genetic distance was estimated between a pair of inbred lines from 6081 markers using Roger's genetic distance (GD) in powerMaker version 3.25 [40]. A relative kinship matrix was calculated between pairs of inbred lines and testers from 6081 SNPs to understand the extent of relatedness using TAS-SELv.5.2.48 [39].

Cluster analysis was performed for the inbred lines and testers based on the genetic distance matrix with the unweighted pair group method of the arithmetic mean clustering algorithm (UPGMA dendrogram) in the PowerMarker version 3.25 [40] and viewed using MEGA, version 6.0 [41]. Population structure analysis was estimated from 6081 SNPs, based on a physical distance of 11 kb between adjacent markers. An admixture model-based clustering method was also used to infer the 33 inbred lines and testers population structure using STRUCTURE, version 2.3.4 [42]. Individuals with a probability of membership $\geq$ 60% were assigned to the same group, while those with $<$ 60% probability memberships in any single group that did not show an ancestry proportion higher than this value was assigned to a "mixed" group [43, 44]. The most probable value of K was estimated using the ad hoc statistic $\Delta K$ [45], depending on the rate of change in the log probability of data between successive K values. Also, principal coordinates analyses (PCoA) was performed based on the 6081 SNPs to distinguish among groups formed by the *Striga* resistant inbred lines and testers using GenAlEx 6.5 software [46].

**Agronomic trait-based diversity assessment.** Agronomic trait-based diversity analysis was carried out using eleven traits, namely grain yield, days to anthesis and silking, anthesis-silking interval, plant height, *Striga* damage ratings and *Striga* emergence counts at 8 and 10 weeks after planting (WAP), ear aspect and ears per plant from the 30 yellow *Striga* maize inbred lines and three testers. Means of the selected traits was first standardized in SAS version 9.4 [47]. Correlation among the different traits was analyzed using statistical analysis software (SAS). The principal component analysis was computed in SAS using the correlation matrix of trait means-centred averaged over environments. Measurements of genetic dissimilarity were then estimated from standardized data using the Euclidian distance matrix, after which has been subjected to cluster the lines and tester using Ward's clustering method [48]. The associations between the agronomic trait-based Euclidean distance matrix and marker-based genetic distance were calculated using GenAlEx 6.5 software [46], and the mantel test was used to determine the significant association between these data sets.

## Results

### Marker-based diversity assessment

Polymorphic information content (PIC) values for 6081 SNP markers used to genotype 30 inbred lines, and three testers ranged from 0.09–0.38 with an average value of 0.28 (Table 1), demonstrating the presence of adequate allelic diversity to establish genetic differences among genotypes. Gene diversity from 6081 SNP markers used to genotype 30 inbred lines and three testers ranged from 0.10 to 0.50 in which 34% of the loci had values between 0.095–0.295 while 66% of them fell in the range of 0.297–0.500 with an average of 0.35. These results indicate high genetic divergence of the *Striga* resistant inbred lines and testers (Table 1).

The number of detected effective alleles (Ne) varied from 1.105 for 60 loci to 2.00 for 92 loci, with a mean of 1.59 per locus. The level of heterozygosity found in about 92% of the inbred lines varied from 0.00 to 0.25, with only 8% of the lines having heterozygosity values

**Table 1. Summary of diversity analysis of 30 *Striga* resistant inbred lines and three testers with varying levels of resistance to *Striga*.**

| Marker information | Values |
|---|---|
| The total number of GBS generated SNP markers | 27,874 |
| Number of markers used for analysis | 6081 |
| Mean Major allele frequency | 0.74 |
| Mean Minor allele frequency | 0.26 |
| Mean Gene Diversity | 0.35 |
| Mean Polymorphic information content (PIC) | 0.28 |

ranging from 0.25 to 1.00. Further analyses revealed that nearly 33% of the markers had observed heterozygosity (Ho) values of 0.00 to 0.05, with only one locus having a heterozygosity value of 1.00, signifying that most loci were fixed. The He value also ranged from 0.012 to 0.50 with a mean of 0.35. The fixation index (F) varied significantly from -1.00 to 1.00, with a mean of 0.66, suggesting higher levels of homozygosity within the lines studied (Table 2).

**Genetic distance and relative kinship analysis.** Analysis of diversity using SNP markers showed wide variation in the genetic distances estimated among inbred lines. The highest mean distances were obtained between L23 and L6 (0.455) and between L25 and L6 (0.453) (S1 Table), indicating that these lines are the most divergent based on SNP markers. In contrast, L23 and L25 exhibited the smallest genetic distance (0.011) though they were not of the same pedigree. In general, lower distances were observed between sister lines with a common pedigree in this study. The genetic distance for pairs of yellow testers was 0.37 for T1 vs T2, 0.38 for T1 vs T3 and 0.40 for T2 vs T3. T3 was distant from the two other testers. The susceptible tester (T3) is a *Striga* susceptible line derived from a bi-parental cross between 9450, a temperate line and KI21, a line from Thailand and therefore expected to have a low frequency of the favourable allele. This possibly contributes to the phenotypic contrast of the test crosses with those of the *Striga* tolerant/resistant inbred lines, which further reflects the higher genetic distance between them.

The genetic distance among *Striga* resistant yellow maize inbred lines and testers with varying level of reactions to *S. hermonthica* ranged from 0.02 to 0.45, with an average of 0.36 (S2 Table). The average genetic distance between *Striga* resistant maize inbred lines on one hand and the susceptible (T3) and resistant (T2) tester was 0.38 and 0.33, respectively. The susceptible tester showed the highest distance from the *Striga* resistant lines compared to the two other testers, while the resistant tester had the lowest genetic distance from the *Striga* resistant maize inbred lines (S2 Table). However, the susceptible tester did not show good hybrid

**Table 2. Genetic parameter for the 6081 codominant SNP markers used to evaluate 30 *Striga hermonthica* resistant maize inbred lines and 3 testers.**

| Genetic parameters | Maximum | Minimum | Mean | SE |
|---|---|---|---|---|
| Na | 2.00 | 1.00 | 1.82 | 0.003 |
| Ne | 2.00 | 1.00 | 1.59 | 0.004 |
| I | 0.69 | 0.00 | 0.52 | 0.002 |
| Ho | 1.00 | 0.00 | 0.12 | 0.001 |
| He | 0.50 | 0.00 | 0.35 | 0.002 |
| uHe | 0.57 | 0.00 | 0.35 | 0.002 |
| F | 1.00 | -1.00 | 0.66 | 0.004 |

SE = Standard error, Ne = no. of effective alleles I = Shannon's information index = $1^*$ Sum (pi $^*$ Ln (pi)), Na = no. of alleles, Ne = Number of effective alleles, Ho = observed heterozygosity, He = expected heterozygosity, uHe = Unbiased heterozygosity and F = fixation index.

**Table 3. Means, maximum, minimum and standard errors of agronomic and morphological traits for each tester evaluated across environments.**

| Testers | YLDIN | YLDUN | DYSK | DYAN | ASI | PL HT |
|---|---|---|---|---|---|---|
| T1 | 3704.88[a] | 4641.89[a] | 57.45[b] | 55.52[b] | 1.93[a] | 158.93[b] |
| T2 | 3814.78[a] | 4803.38[a] | 58.64[a] | 56.56[a] | 2.08[b] | 165.80[a] |
| T3 | 2930.78[b] | 4032.79[b] | 55.96[c] | 54.03[c] | 1.93[a] | 155.65[c] |
| LSD (0.05) | 195.65 | 164.66 | 0.41 | 0.38 | 0.12 | 2.89 |
| Testers | STRRAT1 | STRRAT2 | STRCO1 | STRCO2 | EASP | EPP |
| T1 | 3.08[a] | 4.56[a] | 31.87[a] | 49.27[ab] | 2.81[a] | 0.94[a] |
| T2 | 3.32[b] | 5.08[b] | 27.58[a] | 46.54[a] | 3.03[b] | 0.95[a] |
| T3 | 4.02[c] | 5.81[c] | 37.52[b] | 55.41[b] | 3.13[c] | 0.85[b] |
| LSD (0.05) | 0.18 | 0.24 | 4.5 | 6.5 | 0.09 | 0.03 |

DYSK = Days to 50% silking, DYSAN = Days to 50% anthesis, PLHT = Plant height (cm), STRRAT1 and STRRAT2 = *Striga* damage rating (rating at a scale of 1–9) at 8 and 10 WAP, respectively, STRCO1 and STRCO2 = *Striga* emergence count at 8 and 10 WAP, respectively, EASP = ear aspect (rating at a scale of 1–5), ASI = Anthesis silking interval, EPP = ear per plant and YLDIN and YLDUN = grain yield (kg/ha) under *Striga* infested and free condition.

combinations for grain yield and *Striga* resistance-related traits in this study (Table 3). Furthermore, about 33% of the inbred lines showed above-average GD with T1 and T2, whereas 70% of *Striga* resistant maize inbred lines had above average GD with T3. The number of lines having above-average genetic distance estimates with the testers was 10 for T1, 11 for T2 and 20 for T3. Inbred lines L3, L6, L7, L8, L9, L22, L23, L24, L25, L26, L27, L28, and L29 had above average genetic distance estimates with at least two testers (S2 Table). Of these lines, L8, L22, L25, L27 and L28 had above average genetic distance estimates from at least two of the testers based on both DArTseq SNP markers and agronomic traits recorded under *Striga* infestation (S2 Table). Inbred line L8 was about the most distant line to the three testers based on both GDs. As shown in Fig 1, the genetic distance estimates between the *Striga* resistant lines and the resistant tester were broader in the range of GD values than those between the *Striga*-resistant and susceptible tester, indicating the resistant tester was considered as a suitable tester.

The distribution of pair-wise kinship values for the 30 yellow endosperm *Striga* resistant lines and three testers showed that 55% of the pair-wise kinship values varied from 0.100 to 0.150, with the remaining 45% of the pair-wise kinship values ranging from 0.151 to 0.250 (Fig 2). This indicates that more than 50% of the lines in the entire group are related to each other, whereas the remaining 45% are fairly distantly related to each other.

Cluster analysis separated the 30 inbred lines and three testers into two major groups and five sub-groups with a good match of the inbred lines with their pedigree information. Cluster-I comprised seven inbred lines: L25, L2, L8, L9, L22, L27, and L26. Three of these lines (L25-L27) were derived from TZE COMP5, whereas the remaining four lines were derived from the same biparental crosses. The second cluster comprised 23 inbred lines and all the three testers and separated into five sub-groups (Fig 3). In general, the inbred lines derived from the same genetic background clustered together. Although the three yellow inbred testers originated from different source populations, they grouped together with nearly 79% of the resistant maize inbred lines that share a common parentage, possibly because they share many common alleles that were not associated with reaction to *S. hermonthica*. For instance, the resistance tester (T2) formed a sub-cluster in C-IIc with three inbred lines, and the susceptible tester (T3) also sub-clustered differently (C-IIa) with three inbred lines (Fig 3).

The optimum number of groups K, which best explain the population structure of the lines and testers, was estimated at 2 (K = 2) using the Evanno et al. [44] method (Fig 4A), indicating the presence of two groups. As shown in Fig 4B, Group 1 (C-I) consisted of 11 inbred lines,

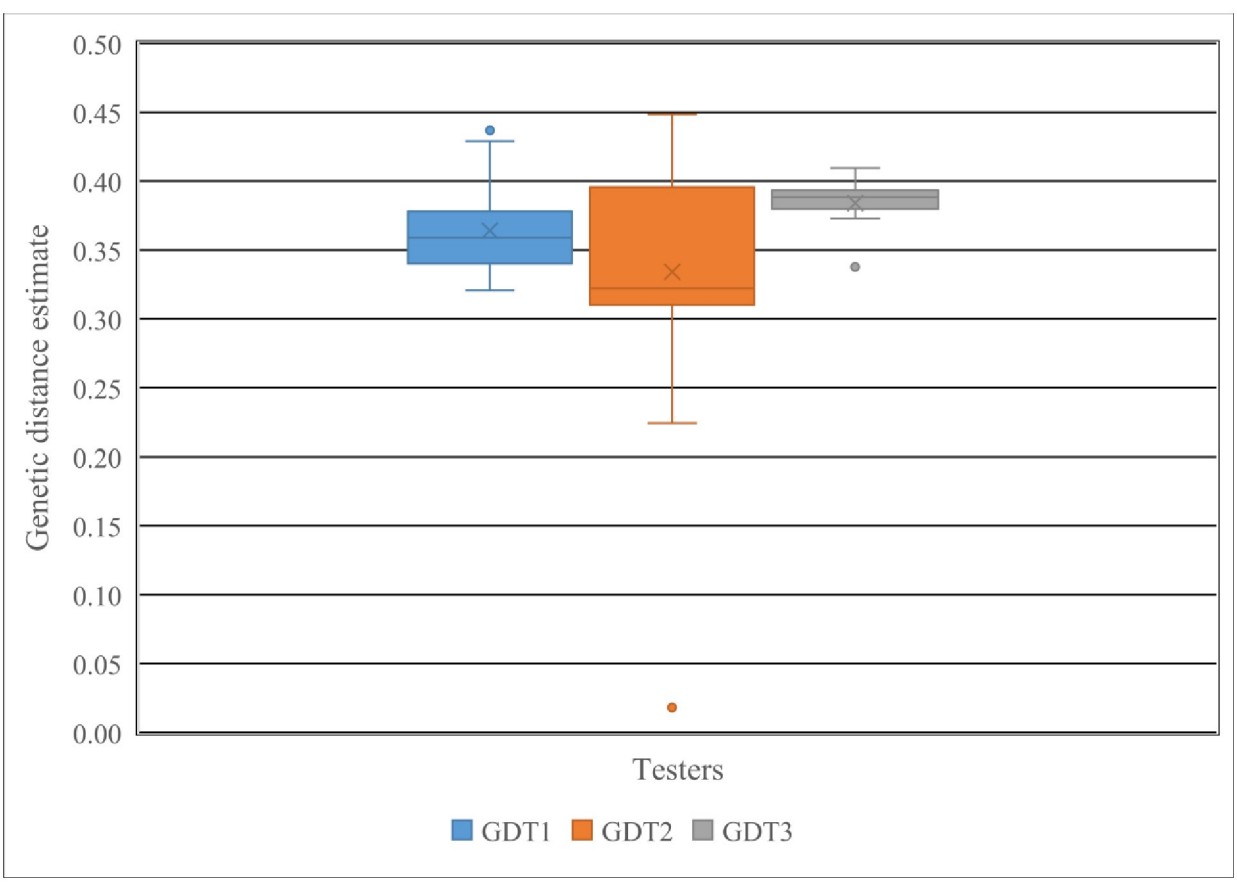

**Fig 1. Genetic distance estimates for combinations of 30 *Striga* resistant yellow endosperm maize inbred lines with each of the three testers (*Striga* tolerant tester (GDT1), *Striga* resistant tester (GDT2) *Striga* susceptible tester (GDT3)) determined using 6081 DArTseq SNPs.**

including the susceptible tester (33%), while group 2 (C-II) composed of 22 inbred lines with the tolerant and resistant testers, representing 67% of the total number of lines. Some lines (L24, L5, L11, L18, and L2) were admixed with the two main group, possibly due to pollen contamination during the breeding process. The admixture model-based population structure

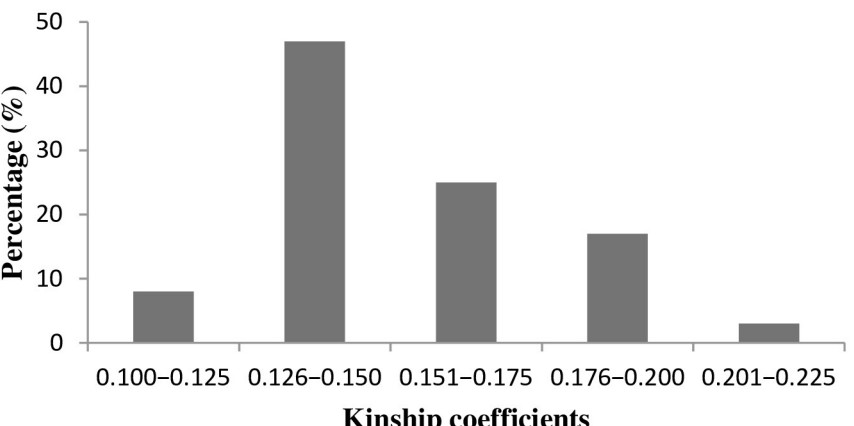

**Fig 2. Distribution of pair-wise relative kinship for thirty inbred lines and three testers with varying level of resistant to *S. hermonthica* calculated using 6081 DArTseq SNPs.**

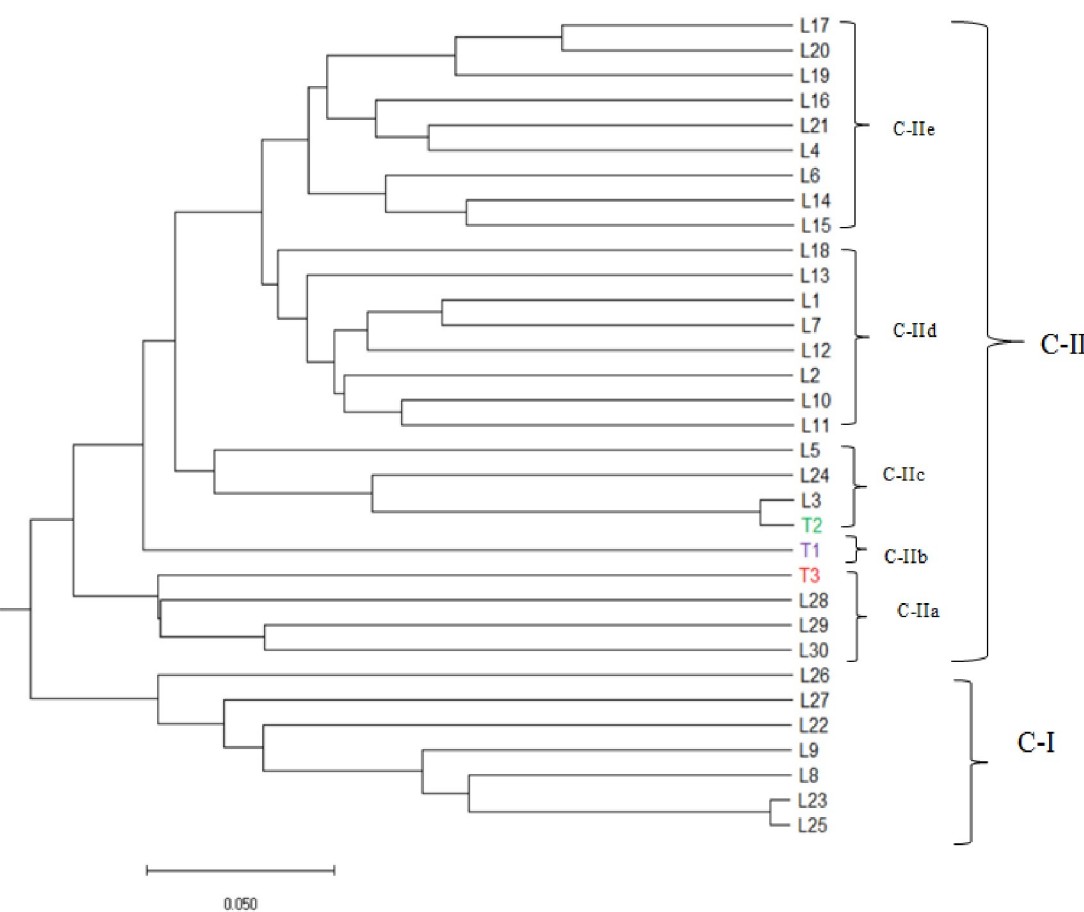

**Fig 3. Dendrogram constructed using the Unweighted Pair Group Method Using Arithmetic Averages (UPGMA) clustering method for 30 *Striga hermonthica* resistant inbred lines and three testers based on Rogers' genetic distance estimates.** Individuals represented in a different colour are testers (Red = Susceptible (T3), Green = Resistant (T2) and Purple = Tolerant testers (T1).

showed a similar pattern as cluster analysis with respect to the number of groups and members of inbred lines in each group. All the seven lines in the C-I group of DArTseq-based cluster analysis were in the C-I group obtained in the admixture model-based population structure. The inbred lines included in group 1 exhibit a broad range in genetic distance estimates.

The principal coordinates analysis (PCoA) also clearly separated the *Striga* resistant inbred lines and testers into two major clusters (Fig 5). Seven *Striga* resistant inbred lines clustered separately in group 1, whereas the remaining 26 clustered in another group (group 2). The first principal component axis explained 29% of the total variance, while the second axis explained 12% of the variance. Inbred lines included in group 1 exhibit a broad range of genetic distance estimates. All the seven lines in the C-I group of the cluster and population structure analysis were in group 1 obtained in the PCoA group. Therefore, the result observed in PCoA also supports the result observed in cluster and structure analysis.

**Agronomic trait-based diversity assessment.** The performances of the three testers for the traits measured across environments are presented in Table 3. The testers in different test-cross combinations were significantly ($P < 0.05$) different for grain yield and other important agronomic traits. T2 had high mean grain yields under infested and non-infested conditions, but these were not significantly different from the means of crosses obtained for T1. However,

A)

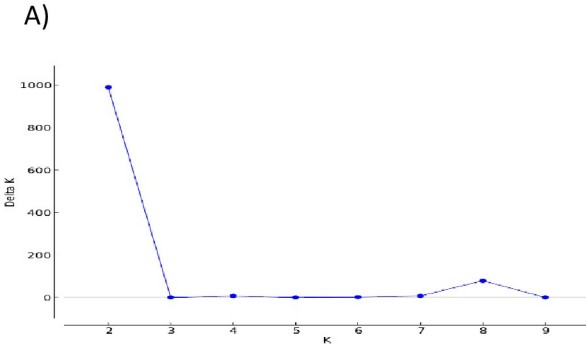

B)

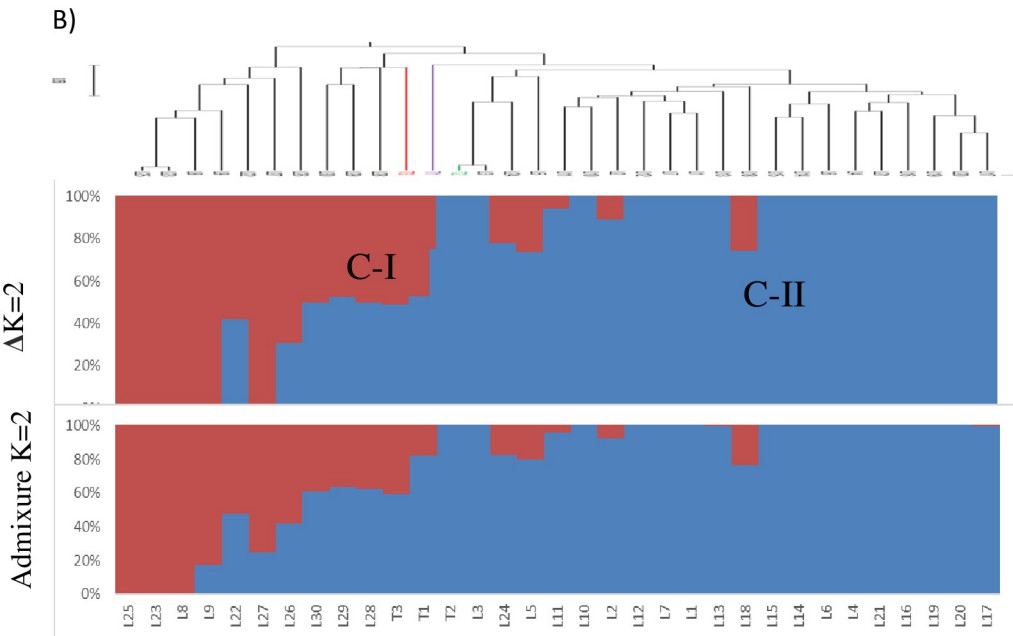

**Fig 4. Population structure of the 30 yellow endosperm maize inbred lines and three testers with varying levels of resistant to *Striga*.**

the mean yields of T2 and T1 were significantly higher than the mean yields of T3 under infested and non-infested conditions. Similar results of non-significantly different performances were obtained between T2 and T1 for STRCO1, STRCO2 and EPP. Except for STRCO2, the values obtained for T2 and T1 were significantly lower values obtained for T3, indicating the superiority of T2 and T1 over T3. For STRRAT1 and STRRAT2, T1 showed significantly lower values than T2, while the values obtained for T2 were also significantly lower than values obtained for T3. From these results, T1 followed by T2 are appropriate testers for grain yield and *Striga*-related traits.

The overall means of the testers in this study showed that the higher the gene frequency of the favourable alleles in a tester, the greater its overall testcross mean. The two testers, T2 and T1, given their resistance/tolerance to *Striga*, were assumed to have higher gene frequencies of favourable alleles for these traits than T3. (Table 3). Simple correlation among agro-morphological and *Striga* related traits was included in S3 Table. Pearson's correlation coefficients

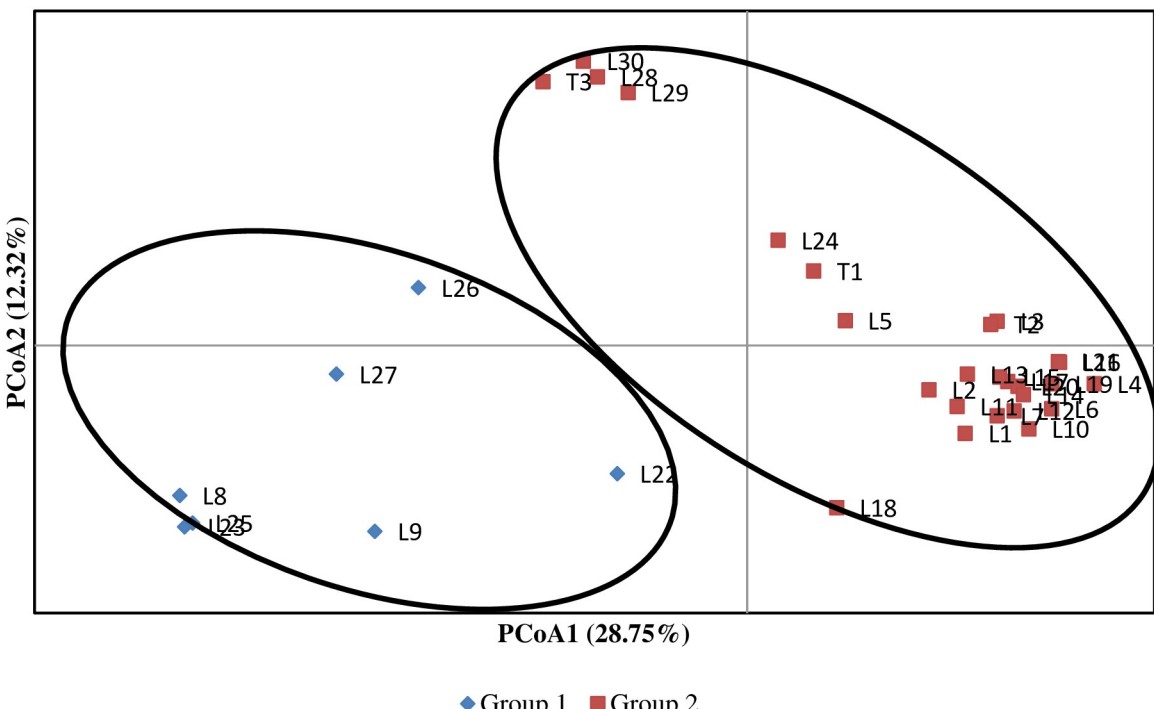

**Fig 5. Plots of value for the first two principal coordinates of the *S. hermonthica* resistant inbred lines and testers with varying levels of *Striga* resistance determined based on clusters identified by population structure analysis.**

among traits included in this study were significant and positive between days to silking (DYSK) and days to anthesis (DYAN) (r = 0.99), between YLD and EPP (r = 0.72), but there were significant and negative between grain yield (YLD) and STRCO1, STRCO2, STRRAT1, STRRAT2 and ear aspect (EASP).

The agronomic traits considered showed a wide variation among the inbred lines and testers in the present study. The greatest mean distance was obtained between L8 and L29 (1.00), followed by L25 and L29 (0.93), indicating that these are the most divergent lines in respect of the traits evaluated (S1 Table). In contrast, pair-wise comparisons between L16 and L7 showed the least divergence (0.00), indicating that these lines were similar for phenotypic traits. The agronomic traits-based distance estimates for the testers was 0.143 for T1 vs T2, 0.491 for T1 vs T3 and 0.493 for T2 vs T3, showing that T3 was again the most divergent among the testers. The number of lines with above-average agronomic trait-based distance estimates in crosses with testers was 14 for T1, 12 for T2 and 11 for T3. *Striga*-resistant inbred lines that showed above-average agronomic trait-based distance estimates in crosses with at least two testers include L5, L8, L12, L13, L17, L22, L25, L27, L29, L30 (S2 Table). Out of all the lines, lines L8, L22, L25, L27, and L28 had above average genetic distance estimates combined with at least two testers based on both DArTseq SNP markers and agronomic traits recorded under *Striga* infestation. As shown in Fig 6, the pairs of the resistant lines and the tolerant tester exhibited.

Cluster analysis of the Euclidean distances calculated from agronomic traits grouped the 30 inbred lines and three testers into two distinct clusters (Fig 7). Cluster I included 14 inbred lines and the susceptible tester (T3), whereas cluster II was composed of 16 inbred lines and the tolerant and resistant testers (T1 and T2). Some of the lines that were clustered in each group using molecular marker showed some inconsistencies with clustering using agronomic trait-based grouping. This result showed that morphological distance is not the primary factor

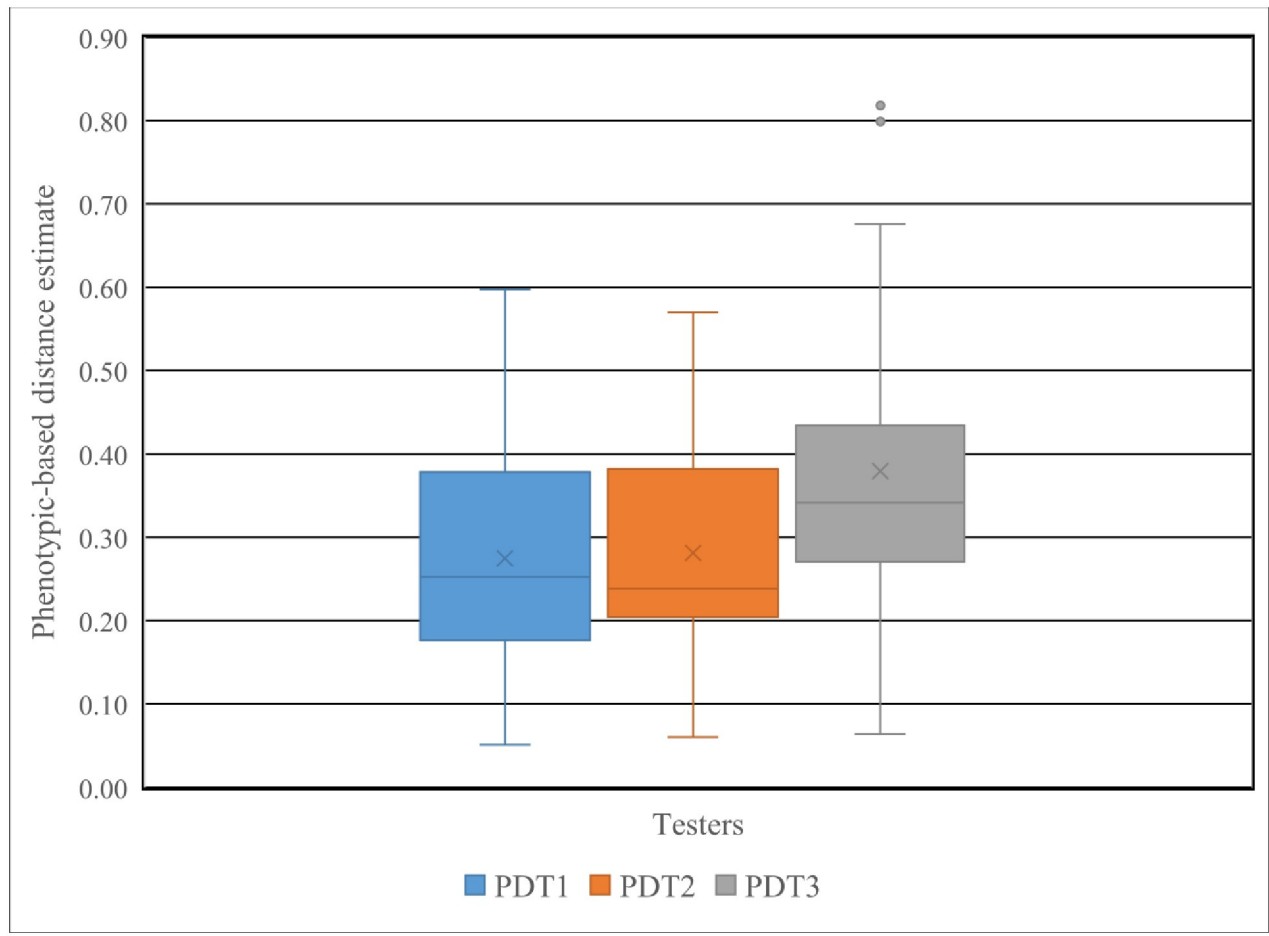

**Fig 6. Phenotype-based distance estimates for combinations of 30 *Striga* resistant yellow endosperm maize inbred lines with each of the three testers (*Striga* tolerant tester (PDT1), *Striga* resistant tester (PDT2) *Striga* susceptible tester (PDT3)) determined using agronomic data measured under *Striga* infestation.**

in separating groups due to the environmental effect determining the performances of the genotype.

**Association between marker-based and agronomic trait-based genetic distance estimates.** Agronomic traits-based distances among the 30 lines and three testers generated based on 11 agronomic traits (S1 Table, below diagonal) were compared with the genetic distance matrices obtained from 6081 SNP data (above diagonal). Regression analysis and Mantel test (S1 Fig) showed a significant (*P* = 0.01) correlation between molecular-based genetic distance and trait-based genetic distance matrices for lines crossed to T1 = 0.362, for lines crossed to T2 = 0.389 and for lines crossed to T3 = 0.360 (S1 Fig). The r-squared (R2) values indicate that between 12.9 and 15.2% of the variation in agronomic traits-based GD is due to a linear function of SNP-based GD variation.

## Discussion

This study was conducted to assess the potential of genetic distance estimates generated using DArTseq SNPs and important agronomic traits measured in hybrids under *S. hermonthica* infestation to identify suitable testers for classifying the combining ability of *Striga* resistant yellow endosperm maize inbred lines. The genetic divergence of the *Striga* resistant inbred

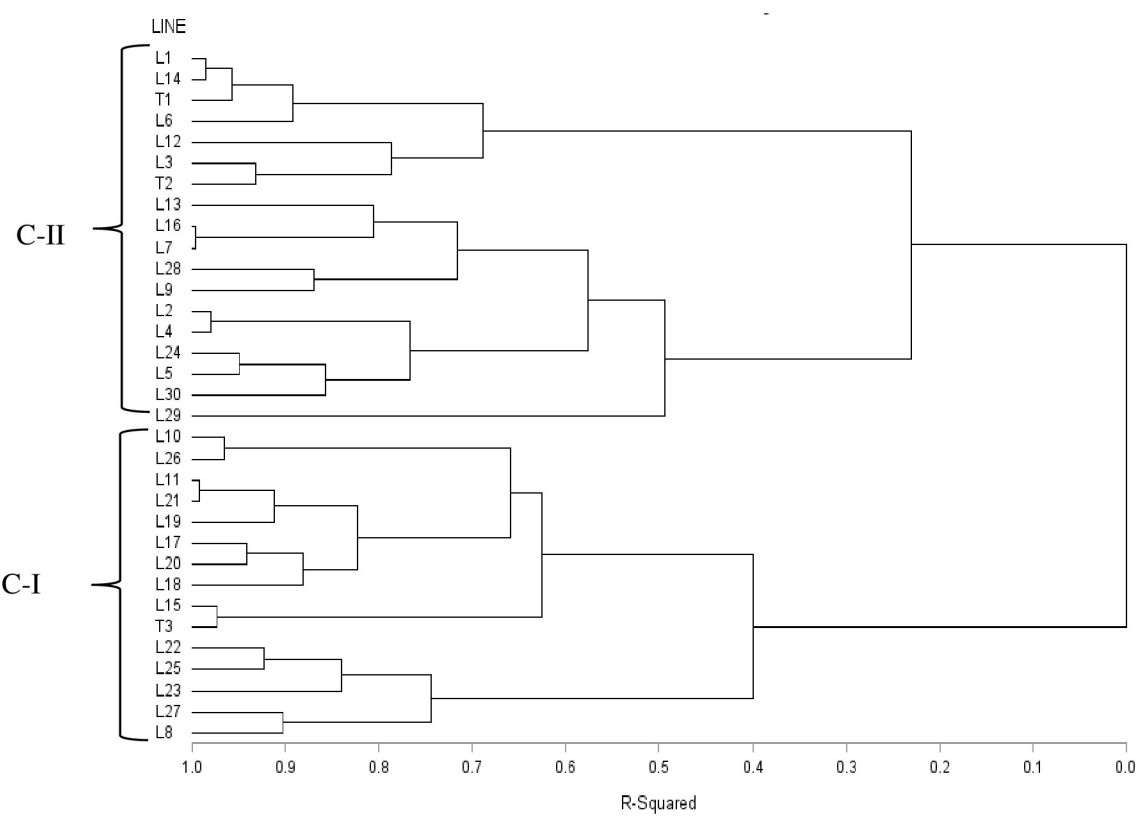

**Fig 7. Dendrogram of 30 yellow maize *Striga* resistant inbred lines and three testers with varying levels of resistance to *Striga* generated by Ward from the dissimilarity matrices.**

lines and testers estimated using PIC value and gene diversity were intermediate to high. The observed PIC and gene diversity values in the present study were somewhat higher than those of Mengesha et al. [49], who reported 0.16 PIC value and gene diversity value of 0.20 in drought-tolerant and *Striga hermonthica* resistant (DSTHR) inbred lines (ACR97TZLComp1) and Lu et al. [43], who reported a mean PIC value of 0.26 and gene diversity of 0.32 from 1034 informative SNP markers used to genotype 770 maize inbred lines.

The genetic distance among breeding materials is considered a key factor in predicting genetic variability among parental combination [50, 51]. The GDs between inbred lines and testers in the present study showed considerable variation, indicating a broad genetic base for the genotypes studied. Information for selecting the best parental combinations to generate new crosses for developing improved maize inbred lines can be obtained using genetic distances estimated from high-density molecular markers [52]. High genetic distance between testers on the one hand and inbred lines being evaluated is one of the major criteria in the determination of an ideal tester when the clusters formed from GD estimates show heterotic relationships [25]. In general, testers allow fewer crosses to be made when new lines are being tested for placement in appropriate heterotic groups. Results showed that less than 6% of the pair-wise comparisons between resistant lines and testers with varying levels of resistant to the parasite had genetic distance estimates of less than 0.30, indicating the presence of adequate genetic diversity among lines and testers that can be exploited to develop high yielding and *Striga* resistant hybrids. Although 70% of pairs of the *Striga* resistant inbred lines with the susceptible tester (T3) had above average genetic distance estimates, T3 exhibited narrow genetic

distance estimates and poor agronomic performance in pairs with resistant lines in the present study.

In contrast, the resistant tester (T2) showed a broader range of genetic distance estimates and superior agronomic performance under *Striga* infestation in pairs with the yellow endosperm *S. hermonthica* resistant inbred lines. The crosses of these inbred lines with the resistant tester also showed fewer *Striga* emergence count and less *Striga* damage rating [37]. In effect, the resistant tester can be regarded as an appropriate tester. However, the choice of tester become difficult when diverse inbred lines are included in the study [53].

The result on kinship coefficient estimation is by far higher than the result of Dao et al. [54], who found pair-wise kinship values close to zero for about 61.3% of 96 INERA and IITA maize inbred lines. Wu et al. [44], in a study involving 367 maize genotypes with known pedigree, reported 94.97% of paired relative kinship having a range of 0.05–0.28; 0.17% having zero kinships, 0.94% with a kinship value of 0.0–0.05 with the remaining 3.92% having a kinship coefficient range of 0.30–0.50. The authors concluded that weak relative kinship existed among the materials studied.

Cluster analysis, principal coordinate analysis and model-based population structure analysis using distance estimates of DArTseq SNPs markers separated the *Striga* resistance yellow endosperm maize inbred lines and testers into distinct groups with a good match of the inbred lines that were grouped together. In general, the inbred lines originating from the same source population clustered together in the present study, consistent with reports in other studies involving tropical maize [28, 29, 55–57, 59]. These results suggest that the lines with different genetic backgrounds that are clustered into different groups could be effectively used as potential parents to develop resistant hybrids with enhanced expression of heterosis. Although the three yellow inbred testers originated from different source populations, they grouped together with nearly 79% of the resistant maize inbred lines that share common parentage possibly because they may share many common alleles that were not located in genomic regions controlling the differential reactions of the lines and testers to *S. hermonthica*. Lines within a group or subgroup in most instances showed a high level of genetic similarity. Consequently, crosses between genetically diverse lines obtained from different groups or subgroups can produce good hybrid than genetically related parent [50, 58]. The lines that are in the same cluster are almost always related by pedigree. This was confirmed by the presence of narrow GD between pairs of such inbred line. On the other hand, some of the lines in the two major groups were not separated into groups in line with their pedigree. Such situations occur mainly due to adaptive selection, markers that are identical in a state not identical by descent, mistaken pedigrees, error in experimentation, or renamed lines in the pedigrees [55].

The use of molecular information combined with those from phenotypic evaluation offers the promise of maximizing genetic variability. Agronomic traits are relatively less reliable and efficient for discrimination among closely related accessions and analysis of genetic relationships than molecular markers. However, genetic diversity assessment based on agronomic traits is fast and straightforward to conduct and may provide preliminary information on genetic diversity among genotypes [59]. Cluster analysis of agronomic trait-based Euclidean distances estimates separated the *Striga* resistance maize inbred lines and the three testers into two distinct groups without due regard to the similarity or difference in the genetic backgrounds of the lines. It was apparent that the crossing of resistant lines with testers having varying levels of resistance to *S. hermonthica* had a significant effect in separating the lines into different groups based on the similarity of performance under parasite pressure. Although the combinations of resistant lines and the susceptible tester showed the largest average agronomic trait-based distance estimate, both the tolerant (T1) and the resistant (T2) testers in crosses with the resistant inbred lines exhibited greater variability in distance estimates and thus could

be regarded as suitable testers. In addition, four inbred lines crossed with the resistant tester viz L6 ×T2, L15 ×T2, L20×T2 and L25 × T2 and two inbred lines crossed to the tolerant tester viz L29×T1 and L30×T1 had high and significant specific combining ability effects (S4 Table). These results indicate that selecting testers that allow accurate identification of lines with superior agronomic performance in hybrid combinations can determine the success of a resistance breeding program. Because many high yielding hybrids with other desirable traits were obtained by crossing the resistant lines with a resistant tester (T2), T2 seems to be a better tester for determining the breeding value of lines in hybrid maize breeding programs for resistance. This result is contrary to the approaches proposed by Smith [21] and Hallauer et al. [13], who support the selection of a tester with a low frequency of favourable alleles for test-cross evaluation to identify lines with greater frequency of favourable alleles.

The significant correlations between marker-based and agronomic trait-based distance estimates of resistant lines with each of the three testers highlight the possibility of using the two types of information for selecting the best parents for resistance breeding. Though all testers crossed with *Striga* resistant inbred lines showed a positive and significant correlation between the genetic distance matrices, the highest correlation (r = 0.389, P = 0.01) was obtained in crosses between the *Striga* resistant inbred lines and resistant tester (T2). The significant correlations between these two sets of data indicate some similarity in the pattern of genetic diversity. Yoseph et al. [59] reported that there was a significant positive correlation between AFLP and SSR (r = 0.39, P < 0.01) based genetic diversity and morphological data r = 0.43, P < 0.01). Aci et al. [60], however, reported negative and significant (r = -0.15, P <0.001) correlation between genetic and agro-morphological distance. These differences could be associated with the type of molecular markers used and stress conditions during evaluation of the materials.

## Conclusion

The pair-wise comparison between most yellow endosperm lines and testers with varying resistance levels revealed adequate genetic diversity among lines and testers. The different DArTseq SNPs based analyses separated the *Striga* resistance yellow endosperm maize inbred lines and testers into distinct groups with an identical set of inbred lines falling within each group. The lines with different genetic backgrounds that are clustered into different groups could be effectively used as potential parents to develop resistant hybrids with greater expression of heterosis. The resistant inbred lines and testers had considerable phenotypic diversity in agronomic traits, possibly due to differences in genetic constitution. The significant correlations between agronomic trait-based and marker-based genetic distance estimates revealed the presence of some level of similarity is genetic diversity patterns of the tested lines. The resistant tester (T2) showed a greater association between the two data sets.

Furthermore, T2 that showed a broader range of marker-based distance estimates and superior agronomic performance in crosses with the resistant inbred lines could be regarded as the most suitable tester. It would also be beneficial to consider the tolerant tester (T1) as an alternative tester as it also attained good hybrid performance following T2. This result can also help other breeders to investigate the relative discriminating ability of testers with different resistance levels in hybrid breeding for resistance to diseases, pests, and other parasitic plants.

## Supporting information

**S1 Table. Molecular genetic distances based on 6081 SNP markers (upper diagonal) and trait-based genetic distance matrices using twelve agronomic traits (lower diagonal).** (DOCX)

**S2 Table. Genetic distance (GD) (Roger's, 1972) and phenotypic distance (PD) between Striga resistant maize inbred lines and testers with varying levels of resistance to *Striga*.**
(DOCX)

**S3 Table. Pearson correlation between traits for testcrosses of yellow endosperm maize and checks under *Striga* infested and non-infested conditions in four environments (n = 92).**
(DOCX)

**S4 Table. Estimates of specific combining ability (SCA) effects of lines and testers evaluated for agronomic and *Striga* resistance traits across four environments under *Striga* infestation.**
(DOCX)

**S1 Fig. Cophenetic correlation between A) *Striga* resistant inbred lines and testers with varying levels of resistance to *S. hermonthica*, B) *Striga* resistant inbred lines and tolerant tester, C) *Striga* resistant inbred lines and resistant tester and D) *Striga* resistant inbred lines and susceptible tester.**
(DOCX)

## Acknowledgments

The authors are grateful to CIMMYT for genotyping and express their appreciation to all staff members of IITA that carried out laboratory analyses and those that participated during planting, data collection, and management of the trial in each location.

## Author Contributions

**Conceptualization:** Abebe Menkir, Melaku Gedil.

**Data curation:** Degife Zebire.

**Formal analysis:** Degife Zebire, Abebe Menkir, Melaku Gedil.

**Funding acquisition:** Abebe Menkir.

**Methodology:** Degife Zebire, Melaku Gedil.

**Project administration:** Abebe Menkir.

**Resources:** Abebe Menkir.

**Supervision:** Abebe Menkir, Victor Adetimirin, Wende Mengesha, Silvestro Meseka, Melaku Gedil.

**Validation:** Abebe Menkir, Victor Adetimirin.

**Writing – original draft:** Degife Zebire.

**Writing – review & editing:** Degife Zebire, Abebe Menkir, Victor Adetimirin, Wende Mengesha, Silvestro Meseka, Melaku Gedil.

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
