## [Decision Letter · Decision Letter 0]

19 Feb 2021

PONE-D-20-38978

Identifying Suitable Tester for Evaluating Striga Resistant Lines using DArTseq Markers and Agronomic Traits

PLOS ONE

Dear Degife Asefa Zebire,

Thank you for submitting your manuscript to PLOS ONE. After careful consideration, we feel that it has merit but does not fully meet PLOS ONE’s publication criteria as it currently stands. Therefore, we invite you to submit a revised version of the manuscript that addresses the points raised during the review process.

We look forward to receiving your revised manuscript.

Kind regards,

Vikas Sharma, Ph.D

Academic Editor

PLOS ONE

Additional Editor Comments:

You will see that reviewers suggesting the thorough revision for your manuscript. Therefore, you are requested to revise the manuscript and address the queries raised by reviewers.

Journal Requirements:

Reviewers' comments:

Reviewer's Responses to Questions

**Comments to the Author**

1. Is the manuscript technically sound, and do the data support the conclusions?

Reviewer #1: Yes

Reviewer #2: Partly

Reviewer #3: Partly

2. Has the statistical analysis been performed appropriately and rigorously? 

Reviewer #1: Yes

Reviewer #2: Yes

Reviewer #3: Yes

3. Have the authors made all data underlying the findings in their manuscript fully available?

Reviewer #1: Yes

Reviewer #2: No

Reviewer #3: Yes

4. Is the manuscript presented in an intelligible fashion and written in standard English?

Reviewer #1: Yes

Reviewer #2: Yes

Reviewer #3: No

5. Review Comments to the Author

Reviewer #1: This study is useful and novel for evaluating the Striga Resistant Lines using suitable tester. Authors used DArTseq

Markers and agronomic traits to identify the suitable tester. The stud is well supported with both sequence based and field based study clearly demarcating the suitable tester for evaluating the Striga Resistant Lines. I recommend this article to publish in your journal. However I have found few spelling mistakes in the article are as following.

There should be a space after genus name S. hermonthica Line 69 and Line 225, Line 291

Tester spelling misspelled in line 243

Furthermore spelling is misspelled in line 465

Reviewer #2: Comments:

1. Please mention details of T1 in the Abstract.

2. In Table-3 as you have compared agronomic traits, add morphological

pictorial parameter to support your data.

3. Reframe the conclusion section for better understanding of the reader.

4. You should perform in vitro or in vivo studies to see If any

adverse effects of these resistant strains for future commercial use.

5. In line 434-435, you should include relevant data.

6. You need to use some more markers along with SNP marker?

Reviewer #3: The manuscript entitled “Identifying Suitable Tester for Evaluating Striga Resistant Lines using DArTseq Markers and Agronomic Traits”, has been extensively reviewed by me and found that Authors have developed Tester which has resistance to Striga. Manuscript is good but there are major concerns in the study which needs to be addressed. There are more typological errors in the whole document and needs to be improved in the revision. Manuscript needs to be improved rigorously and based on my analysis I recommend major revision. I consider this manuscript only when all the queries will be answered thoroughly.

Minor Comments:

1. Maize (Zea mays L; 2n = 2x = 20) is one of the most important cereal crops worldwide by 60 providing food, feed and bioenergy [1, 2]…..correct the sentences.

2. 69 affected areas [9]. Inbred lines that combine high yield with resistance to S.hermonthica can be…..there should be space.

3. 84 Smith [21] and Hallauer et al. [13] concluded that a homozygous recessive inbred line with a low 85 frequency of desirable dominant genes and favourable alleles would be the most effective tester to 86 identify new inbred lines with high frequency of favourable alleles in testcrosses……lines are not clear improve this.

4. 88 [22, 13]. Mwimali et al. [23] found two Busseola fusca resistant single-crosses as desirable testers…scientific name should be italics.

5. 121 The present studies were, therefore, conducted to characterize Striga resistan…..it should be……The present study is….

6. 138 for DNA extraction. Leaves were collected from 4 to 15 leaves of each line and tester and then 139 bulked. The samples…..my question is that why there is need to bulk the DNA.

7. 179 first standardized in SAS version 9.4 [47]. Correlation among the different traits were analyzed… It should be traits was

8. The principal component analysis was computed in SAS using the correlation

181 matrix of trait means-centered averaged over environments….write the full form of SAS software.

9. 225 level of reactions to S.hermonthica ranged from 0.02 to 0.45 with an average of 0.36….scientific name should be with space.

10. 295 The performances of the three tester for the traits measured across environments are presented in 296 Table 3…language written is very simple inprove the sentence.

11. 164 Population structure analysis was performed from 6081 SNPs and a physical distance of 11 kb. 165 between adjacent markers…sentences is incomplete.

Major Commnets:

12. Why authors have not used SSRs markers as they are also widely distributed in the genome?

13. 133 tolerant and susceptible reactions to the parasite, under Striga infested and Striga noninfested 134 conditions at Abuja and Mokwa in Nigeria for two years. …Authors have used only two …I always see it should be 3 years….I thinks there will be biased results, chances of association of false SNPs are there….explain it?

14. The authors have used only 30 samples ….which I think it’s the only major drawback of the manuscript….at least 96 is required to find the marker trait association….. add more samples,,,,because with 30 samples you will get false results.

15. The samples were sent to 145 the Diversity Array technology company [38]. All protocols including Library construction, 146 sequencing, and SNP calling were performed at the Diversity Arrays facility, Canberra, Australia…..write all the protocols in the methods…

16. 192 genotypes. Gene diversity from 6081 SNP markers used to genotype 30 inbred lines and three 193 testers ranged from 0.10 to 0.50 in which 34% of the loci had values between 0.095-0.295 while 194 66 % of them fell in the range of 0.297-0.500 with an average of 0.35. These results indicate high 195 genetic divergence of the Striga resistant inbred lines and testers..Gene diversity value is very low….which again due to less number of samples.

17. The level of heterozygosity found in about 92% of the inbred lines

200 varied from 0.00 to 0.25 with only 8% of the lines having heterozygosity values ranging from 0.25 201 to 1.00. Further analyses revealed that nearly 33% of the markers had observed heterozygosity 202 (Ho) values of 0.00 to 0.05 with only one locus having a heterozygosity value of 1.00, signifying 203 that most of the loci were fixed. The He value also ranged from 0.012 to 0.50 with a mean of 0.35. 204 The fixation index (F) varied significantly from -1.00 to 1.00 with a mean of 0.66, suggesting that205 the presence of higher levels of homozygosity within the lines that were studied….value of heterozygosity is also very low….which showed your samples are rich in homozygosity.

18. Legend to figures come in the end,,,,improve it.

19. There are lots of typological errors in the whole manuscript, correct them?

20. Discussion and conclusion needs to be strengthened?

21. In reference section scientific names needs to be italics, authors have to be serious while submit revise manuscript?

6. PLOS authors have the option to publish the peer review history of their article (what does this mean?). If published, this will include your full peer review and any attached files.

Reviewer #1: **Yes: **Dr Vikrant Jaryan

Reviewer #2: No

Reviewer #3: No

---

## [Author Response · Author response to Decision Letter 0]

9 Apr 2021

Reviewer #1: This study is useful and novel for evaluating the Striga Resistant Lines using suitable tester. Authors used DArTseq

Markers and agronomic traits to identify the suitable tester. The stud is well supported with both sequences based and field-based study clearly demarcating the suitable tester for evaluating the Striga Resistant Lines. I recommend this article to publish in your journal. However, I have found few spelling mistakes in the article are as following.

Question

There should be a space after genus name S. hermonthica Line 69 and Line 225, Line 291

Tester spelling misspelled in line 243

Furthermore spelling is misspelled in line 465

Response: All issues concerning spelling errors and space between phrases and words has been checked.

Reviewer #2: Comments:

1. Please mention details of T1 in the Abstract. 

Response: The details of T1 has been introduced as tolerant tester

2. In Table-3 as you have compared agronomic traits, add morphological

pictorial parameter to support your data.

Response: Some morphological parameters are included in the result section.

3. Reframe the conclusion section for better understanding of the reader.

Response: Some modifications has been done in the conclusion section

4. You should perform in vitro or in vivo studies to see If any adverse effects of these resistant strains for future commercial use.

Response: Studies showed there is a positive association between in vivo and in vitro studies on resistant strains the parasite for Striga attachment count. E.g., Assessment of reactions of diverse maize inbred lines to Striga hermonthica (Del.) Benth (Menkir 2006)

5. In line 434-435, you should include relevant data. 

Response: List of crosses with high SCA in the two testers have been included.

6. You need to use some more markers along with SNP marker? 

Response: It is beyond this study

 

Reviewer #3: 

The manuscript entitled “Identifying Suitable Tester for Evaluating Striga Resistant Lines using DArTseq Markers and Agronomic Traits”, has been extensively reviewed by me and found that Authors have developed Tester which has resistance to Striga. Manuscript is good but there are major concerns in the study which needs to be addressed. There are more typological errors in the whole document and needs to be improved in the revision. Manuscript needs to be improved rigorously and based on my analysis I recommend major revision. I consider this manuscript only when all the queries will be answered thoroughly.

Minor Comments:

1. Maize (Zea mays L; 2n = 2x = 20) is one of the most important cereal crops worldwide by 60 providing food, feed and bioenergy [1, 2]…..correct the sentences. 

Response: the sentence under this statement has been improved 

2. 69 affected areas [9]. Inbred lines that combine high yield with resistance to S.hermonthica can be…..there should be space.

Response: space has been provided between S. and hermonthica

3. 84 Smith [21] and Hallauer et al. [13] concluded that a homozygous recessive inbred line with a low 85 frequency of desirable dominant genes and favourable alleles would be the most effective tester to 86 identify new inbred lines with high frequency of favourable alleles in testcrosses……lines are not clear improve this. 

Response: the sentence under this statement has been improved 

4. 88 [22, 13]. Mwimali et al. [23] found two Busseola fusca resistant single-crosses as desirable testers…scientific name should be italics. 

Response: the scientific name of Busseola fusca has been Italicized in L88. 

5. 121 The present studies were, therefore, conducted to characterize Striga resistan…..it should be……The present study is…. 

Response: the statement under L121 has been corrected 

6. 138 for DNA extraction. Leaves were collected from 4 to 15 leaves of each line and tester and then 139 bulked. The samples…..my question is that why there is need to bulk the DNA. 

Response: The DNA was not bulked rather the leaf samples collected from each inbred lines and testers were bulked. To avoid confusion, we rephrased the statement.

7. 179 first standardized in SAS version 9.4 [47]. Correlation among the different traits were analyzed… It should be traits was… 

Response: the statement under L185 has been corrected 

8. The principal component analysis was computed in SAS using the correlation

181 matrix of trait means-centered averaged over environments….write the full form of SAS software. 

Response: SAS is written in full form in L187.

9. 225 level of reactions to S.hermonthica ranged from 0.02 to 0.45 with an average of 0.36….scientific name should be with space. 

Response: Space has been provided between S.hermonthica as S. hermonthica

10. 295 The performances of the three tester for the traits measured across environments are presented in 296 Table 3…language written is very simple inprove the sentence.

Response: The sentence in this line has improved by adding some technical words and phrases.

11. 164 Population structure analysis was performed from 6081 SNPs and a physical distance of 11 kb. 165 between adjacent markers…sentences is incomplete. 

Response: The sentence has completed by rephrasing the whole statement.

Major Comments:

12. Why authors have not used SSRs markers as they are also widely distributed in the genome? Response: This was due to a large increase in throughput, independence of DNA sequence information and low cost. The cost advantage compared to marker-by-marker methods is also dramatic.

13. 133 tolerant and susceptible reactions to the parasite, under Striga infested and Striga noninfested 134 conditions at Abuja and Mokwa in Nigeria for two years. …Authors have used only two …I always see it should be 3 years….I thinks there will be biased results, chances of association of false SNPs are there….explain it? 

Response: We conducted this research in two locations and two years giving a total of four environments, so the chance of getting biased results which will lead to association of false SNPs would be low.

14. The authors have used only 30 samples ….which I think it’s the only major drawback of the manuscript….at least 96 is required to find the marker trait association….. add more samples,,,,because with 30 samples you will get false results. 

Response: The main objective of this study was to identify suitable tester/s to evaluate Striga resistant inbred lines. Though the sample numbers were limited, we used DArTseq SNP markers to see the usefulness of these markers in achieving the main objective in addition to determining the extent of genetic diversity.

15. The samples were sent to 145 the Diversity Array technology company [38]. All protocols including Library construction, 146 sequencing, and SNP calling were performed at the Diversity Arrays facility, Canberra, Australia…..write all the protocols in the methods… 

Response: The protocols performed in DArT for library construction, SNP calling and sequencing are briefly explained from L147-152.

16. 192 genotypes. Gene diversity from 6081 SNP markers used to genotype 30 inbred lines and three 193 testers ranged from 0.10 to 0.50 in which 34% of the loci had values between 0.095-0.295 while 194 66 % of them fell in the range of 0.297-0.500 with an average of 0.35. These results indicate high 195 genetic divergence of the Striga resistant inbred lines and testers..Gene diversity value is very low….which again due to less number of samples. 

Response: The sample numbers were small but the association of marker information and agronomic trait information provided us good result for future breeding programmes and also the main objective of this study was to identify effective tester/s to evaluate Striga resistant inbred lines in addition to studying the extent of genetic diversity.

17. The level of heterozygosity found in about 92% of the inbred lines

200 varied from 0.00 to 0.25 with only 8% of the lines having heterozygosity values ranging from 0.25 201 to 1.00. Further analyses revealed that nearly 33% of the markers had observed heterozygosity 202 (Ho) values of 0.00 to 0.05 with only one locus having a heterozygosity value of 1.00, signifying 203 that most of the loci were fixed. The He value also ranged from 0.012 to 0.50 with a mean of 0.35. 204 The fixation index (F) varied significantly from -1.00 to 1.00 with a mean of 0.66, suggesting that205 the presence of higher levels of homozygosity within the lines that were studied….value of heterozygosity is also very low…. which showed your samples are rich in homozygosity. 

Response: Yes, the inbred lines are advanced lines generated from several generations of selfing.

18. Legend to figures come in the end,,,,improve it. 

Response: The description of the legends for Fig 1, 2 and 5 has been improved.

19. There are lots of typological errors in the whole manuscript, correct them? 

Response: All typological errors are carefully checked and grammatical corrections were done by colleague from English speaking country.

20. Discussion and conclusion needs to be strengthened?

Response: The discussion and conclusion section of the manuscript has be strengthened by adding more information and explanations.

21. In reference section scientific names needs to be italics, authors have to be serious while submit revise manuscript? 

Response: All scientific name inside the main body and reference section has been italicized

---

## [Decision Letter · Decision Letter 1]

24 May 2021

PONE-D-20-38978R1

Identifying Suitable Tester for Evaluating Striga Resistant Lines using DArTseq Markers and Agronomic Traits

PLOS ONE

Dear Dr. Abebe Menkir

Thank you for submitting your manuscript to PLOS ONE. After careful consideration, we feel that it has merit but does not fully meet PLOS ONE’s publication criteria as it currently stands. Therefore, we invite you to submit a revised version of the manuscript that addresses the points raised during the review process.

We look forward to receiving your revised manuscript.

Kind regards,

Vikas Sharma, Ph.D

Academic Editor

PLOS ONE

Journal Requirements:

Additional Editor Comments (if provided):

Although, manuscript is complete now but due to some doubts and language errors it require one more revision. Please check the below mentioned points carefully:

Line 28-30 revise the sentence to make it clearer it is confusing or beak it into two.

Line 31: "marker-based distance estimates (r= 0.389, P= 0.01) was observed for when T2 has paired" remove grammatical errors.

Key Words: "mantel test" should be " Mantel test".

Lines 62-65: which 81 lines please clear, also revise sentence.

Lines 116-118:  "generate 90 test crosses, which were evaluated along with two checks" are these 90  samples were evaluated???  

line 163: "from the 30 yellow Striga maize inbred lines and three testers" only 30 maize imbred lines and three testers were evaluated?????

both these statements in lines 116-118 and line 163 are conflicting please check where is the error and make a correlation in study.

you are using the term "DArT-seq" however, standard term is "DArT"  check care and use appropriate term.

Language require careful revision throughout manuscript, therefore, revise the manuscript  thoroughly.

Reviewers' comments:

Reviewer's Responses to Questions

**Comments to the Author**

1. If the authors have adequately addressed your comments raised in a previous round of review and you feel that this manuscript is now acceptable for publication, you may indicate that here to bypass the “Comments to the Author” section, enter your conflict of interest statement in the “Confidential to Editor” section, and submit your "Accept" recommendation.

Reviewer #1: All comments have been addressed

Reviewer #2: All comments have been addressed

Reviewer #3: All comments have been addressed

2. Is the manuscript technically sound, and do the data support the conclusions?

Reviewer #1: Yes

Reviewer #2: Partly

Reviewer #3: Yes

3. Has the statistical analysis been performed appropriately and rigorously? 

Reviewer #1: Yes

Reviewer #2: Yes

Reviewer #3: Yes

4. Have the authors made all data underlying the findings in their manuscript fully available?

Reviewer #1: Yes

Reviewer #2: Yes

Reviewer #3: Yes

5. Is the manuscript presented in an intelligible fashion and written in standard English?

Reviewer #1: Yes

Reviewer #2: Yes

Reviewer #3: Yes

6. Review Comments to the Author

Reviewer #1: Although authors have addressed all the comments but still I find some spelling and punctuation mistakes in the manuscript. The manuscript is improved by incorporating all the comments by various reviewer. I like to point these mistakes which is yet to be correct in the manuscript.

Line 21 Striga should be in italics.

Line 52 & Line 210 S.hermonthica there should be space after genus name

Line 71 Busseola fusca the name of species should be in italics

Line 123 -80°c here symbol of celcius in capital (°C)

Reviewer #2: The author have addressed all the raised issues and incorporated changes wherever suggested in previous review. So article can be accepted for publication, however, it is advised to proofread article to remove linguistic errors. Further advised to add more markers along with SNP markers for future studies.

Reviewer #3: The manuscript entitled Identifying Suitable Tester for Evaluating Striga Resistant Lines using DArTseq Markers and Agronomic Traits has been satisfactorily revised by the authors and can be accepted for publication. There are still minor mistakes and typological errors which can be corrected at the time of proofreading.

7. PLOS authors have the option to publish the peer review history of their article (what does this mean?). If published, this will include your full peer review and any attached files.

Reviewer #1: No

Reviewer #2: No

Reviewer #3: No

---

## [Author Response · Author response to Decision Letter 1]

1 Jun 2021

Dear Editor, 

We have carefully revised the first version of our manuscript and made some changes based on the reviewer’s recommendation. The comments were very useful to revise our manuscript. 

We are glad to send our manuscript titled “Identifying Suitable Tester for Evaluating Striga Resistant Lines using DArTseq Markers and Agronomic Traits” for publication in PLOS ONE. 

Here are the responses

Editor feedback

Line 28-30 revise the sentence to make it clearer it is confusing or beak it into two.

Response: The sentence broken into two and articulated in a clear manner.

Line 31: "marker-based distance estimates (r= 0.389, P= 0.01) was observed for when T2 has paired" remove grammatical errors.

Response: The grammatical errors in line 31 have been removed by removing unnecessary words and adding appropriate words and prepositions.

Key Words: "mantel test" should be " Mantel test".

Response: it has been changed according to the editor comment.

Lines 62-65: which 81 lines please clear, also revise sentence.

Response: The statement under the indicated lines has been revised by removing inappropriate words, articles and phrase.

Lines 116-118: "generate 90 test crosses, which were evaluated along with two checks" are these 90 samples were evaluated??? 

Response: Yes, these 90 samples are testcrosses obtained from crossing of 30 Striga resistant inbred lines and three testers with varying levels of resistance to Striga hermonthica for assessing the agronomic performance. A phrase has been added to make the information clear and why this is done in line 118.

line 163: "from the 30 yellow Striga maize inbred lines and three testers" only 30 maize inbred lines and three testers were evaluated?????

Response: The 30 yellow Striga maize inbred lines and three testers were genotyped for assessing genetic diversity

both these statements in lines 116-118 and line 163 are conflicting please check where is the error and make a correlation in study.

Response: Correction has made on line 118 to avoid confusion.

 You are using the term "DArT-seq" however, standard term is "DArT" check care and use appropriate term.

Response: Regarding DArT-seq, we used DArTseq (NGS based genotyping) is a newer version of the older DArT (hybridization based) platform. So we have changed the form accordingly.

Language requires careful revision throughout manuscript, therefore, revise the manuscript thoroughly.

Response: We have revised the grammatical errors as much as we can throughout the manuscript.

Reviewer’s Comments 

Reviewer #1: 

Line 21 Striga should be in italics.

Response: The word Striga is in italic form

Line 52 & Line 210 S.hermonthica there should be space after genus name

Response: A space has been provided between S.hermonthica throughout the manuscript as S. hermonthica

 Line 71 Busseola fusca the name of species should be in italics

Response: The species name of Busseola fusca is italized in line 71

Line 123 -80°c here symbol of celcius in capital (°C)

Response: The symbol of degree Celsius in line 121, C is capitalized and shown in the track change

Reviewer #2 

The author have addressed all the raised issues and incorporated changes wherever suggested in previous review. So article can be accepted for publication, however, it is advised to proofread article to remove linguistic errors. Further advised to add more markers along with SNP markers for future studies.

Response: though there is no much comments for reviewer 2, we will see the comparative advantage of SNP markers in our future study

Reviewer #3: 

minor mistakes and typological errors which can be corrected at the time of proofreading.

Response: Most typological errors have been addressed as much as possible in this revised version.

---

## [Editor Report · Decision Letter 2]

7 Jun 2021

Identifying Suitable Tester for Evaluating Striga Resistant Lines using DArTseq Markers and Agronomic Traits

PONE-D-20-38978R2

Dear Dr. Degife Zebire,

We’re pleased to inform you that your manuscript has been judged scientifically suitable for publication and will be formally accepted for publication once it meets all outstanding technical requirements.

Kind regards,

Vikas Sharma, Ph.D

Academic Editor

PLOS ONE

Additional Editor Comments (optional):

The authors have addressed all the raised issues. Manuscript can be accepted now in its present form.
---

## [Editor Report · Acceptance letter]

9 Jun 2021

PONE-D-20-38978R2 

Identifying Suitable Tester for Evaluating *Striga* Resistant Lines using DArTseq Markers and Agronomic Traits 

Dear Dr. Zebire:

I'm pleased to inform you that your manuscript has been deemed suitable for publication in PLOS ONE. Congratulations! Your manuscript is now with our production department. 

Kind regards, 

on behalf of

Dr. Vikas Sharma 

Academic Editor

PLOS ONE